# CCL4 Affects Eosinophil Survival via the Shedding of the MUC1 N-Terminal Domain in Airway Inflammation

**DOI:** 10.3390/cells14010033

**Published:** 2024-12-31

**Authors:** Yoshiki Kobayashi, Chu Hong Hanh, Naoto Yagi, Nhi Kieu Thi Le, Yasutaka Yun, Akihiro Shimamura, Kenta Fukui, Akitoshi Mitani, Kensuke Suzuki, Akira Kanda, Hiroshi Iwai

**Affiliations:** 1Airway Disease Section, Department of Otorhinolaryngology, Kansai Medical University, Hirakata, Osaka 573-1010, Japan; honghanh87.hmu@gmail.com (C.H.H.); kieunhiyds2015@gmail.com (N.K.T.L.); yunys@hirakata.kmu.ac.jp (Y.Y.); shima.8031@gmail.com (A.S.); ken0630bbt@gmail.com (K.F.); mitaniak@hirakata.kmu.ac.jp (A.M.); suzukken@hirakata.kmu.ac.jp (K.S.); akanda@hirakata.kmu.ac.jp (A.K.); iwai@hirakata.kmu.ac.jp (H.I.); 2Allergy Center, Kansai Medical University Hospital, Osaka 573-1010, Japan; 3Third Department of Internal Medicine, Kansai Medical University, Osaka 573-1010, Japan; naoto.y0616@gmail.com

**Keywords:** airway epithelial cell, bronchial asthma, CCL4, chronic rhinosinusitis with nasal polyps, eosinophilic chronic rhinosinusitis, mucin glycoprotein 1

## Abstract

Eosinophilic chronic rhinosinusitis (ECRS), a CRS with nasal polyps (CRSwNP), is characterized by eosinophilic infiltration with type 2 inflammation and is highly associated with bronchial asthma. Intractable ECRS with poorly controlled asthma is recognized as a difficult-to-treat eosinophilic airway inflammation. Although eosinophils are activated and coincubation with airway epithelial cells prolongs their survival, the interaction mechanism between eosinophils and epithelial cells is unclear. This study examined the effect of eosinophils on mucin glycoprotein 1 (MUC1), a member of membrane-bound mucin, in the airway epithelial cells, to elucidate the mechanisms of the eosinophil–airway epithelial cell interaction. Nasal polyp samples from patients with CRSwNP and BEAS-2B airway epithelial cells, coincubated with purified eosinophils, were stained with two MUC1 antibodies. To confirm the involvement of CCL4, an anti-CCL4 neutralizing antibody or recombinant CCL4 was used as needed. The immunofluorescence results revealed a negative correlation between the expression of full-length MUC1 and eosinophil count in nasal polyps. In BEAS-2B coincubated with eosinophils, full-length MUC1, but not the C-terminal domain, was reduced, and eosinophil survival was prolonged, which was concomitant with CCL4 increase, whereas the anti-CCL4 neutralizing antibody decreased these reactions. The survival of eosinophils that contacted recombinant MUC1 without the N-terminal domain was prolonged, and recombinant CCL4 increased the expression of metalloproteases. Increased CCL4 induces the contact between eosinophils and airway epithelial cells by shedding the MUC1 N-terminal domain and enhances eosinophil survival in eosinophilic airway inflammation. This novel mechanism may be a therapeutic target for difficult-to-treat eosinophilic airway inflammation.

## 1. Introduction

Eosinophilic chronic rhinosinusitis (ECRS), a subtype of chronic rhinosinusitis with nasal polyps (CRSwNP), is characterized by eosinophilic infiltration with type 2 inflammation and is highly associated with bronchial asthma [1,2]. Intractable ECRS is complicated by poorly controlled asthma and is recognized as a difficult-to-treat eosinophilic airway inflammation that responds poorly to corticosteroids [3,4,5].

The survival of activated eosinophils is prolonged when they are coincubated with airway epithelial cells [6]. Previously, we found that corticosteroid sensitivity was significantly reduced in airway epithelial cells obtained from patients with ECRS complicated by severe asthma and in BEAS-2B, a human bronchial epithelial cell line coincubated with purified eosinophils [5,6]. This finding indicates that both eosinophils and airway epithelial cells affect each other in their functions. However, the mechanisms of their interaction have not been elucidated.

Thirteen mucin glycoproteins (encoded by MUC), which are major macromolecular mucus components, were found in the airway [7]. Of these 13 mucins, 7 are predominantly located in the airway. MUC1, MUC4, MUC16, and MUC20 are membrane-associated types, MUC5AC and MUC5B are gel-forming types, and MUC7 is secreted and a nongel-forming type [8]. A recent study showed that in nasal epithelial cells, MUC5AC and MUC5B, major mucins in the mucosa of patients with CRS, are induced by IL-33, the levels of which are increased in ECRS [9]. Furthermore, MUC5AC and MUC5B could be involved in forming bacteria biofilms [10,11]. Conversely, MUC1, a subtype of membrane-bound mucins expressed in the respiratory tract, inhibits bacteria- or virus-induced airway inflammation [12,13]. The tethered MUC1 protein, which contains an N-terminal extracellular domain and a C-terminal domain acts as a membrane receptor. The N-terminal extracellular domain on airway epithelial cells acts as a contact point to pathogens or adhesion molecules in other cells [14,15,16]. The C-terminal domain consists of an extracellular stem, a single-pass transmembrane domain, and a cytoplasmic tail (CT), which can modulate multiple intracellular signals [17].

MUC1-CT exerts an anti-inflammatory effect on the airway [12,18,19,20]. Interestingly, in patients with severe asthma or CRSwNP, MUC1-CT downregulation reduces corticosteroid responses because of the impaired glucocorticoid receptor (GR) translocation into the nucleus [21,22]. Asthma is exacerbated in patients with lower MUC1 levels [23]. Although MUC1 was found to be associated with neutrophil inflammation in asthma [24], the association of MUC1 with eosinophilic airway inflammation is unclear. We hypothesized that eosinophils first contact the N-terminal extracellular domain of MUC1 on airway epithelial cells and then exert some influence on airway inflammation. This study focused on the effect of eosinophils on MUC1 in the airway epithelial cells to elucidate the mechanisms of eosinophil–airway epithelial cell interaction, which will help to find a new therapeutic target for difficult-to-treat eosinophilic airway inflammation.

## 2. Materials and Methods

### 2.1. Cell Preparation

Human eosinophils were separated with high purity (>98%) from peripheral blood obtained from healthy individuals with slightly increased eosinophils (approximately 300–500/μL) using an eosinophil isolation kit with a MACS system (Miltenyi Biotec, Bergish Gladbach, Germany). BEAS-2B (European Collection of Authenticated Cell Culture, Salisbury, UK), human bronchial epithelial cells, were coincubated with purified eosinophils or recombinant human CCL4 (Abcam, Cambridge, UK) or anti-CCL4 neutralizing antibody (R&D Systems, Minneapolis, MN, USA), as appropriate. Briefly, BEAS-2B cells seeded in a cell culture plate one day before were stimulated with recombinant human CCL4 overnight (for 20–24 h) or were coincubated overnight with purified eosinophils in the presence or absence of anti-CCL4 antibody overnight.

### 2.2. Quantitative RT-PCR

RNA was extracted from BEAS-2B using the RNeasy Mini Kit (Qiagen, Hilden, Germany). Then, the reverse-transcription of cDNA using Perfect Real Time (Takara, Shiga, Japan), and qPCR using a QuantiTect SYBR Green PCR kit (Qiagen) on a Rotor-Gene Q (Qiagen) were performed. The relative gene expressions of the N-terminus domains of MUC1, ADAM17 and MMP14 were analyzed using the 2^−ΔΔCt^ method. GAPDH was used as a reference. The amplification primers (5′–3′) for analysis were as follows: MUC1 [TGC TGC TCC TCA CAG TGC TTA C (forward), TCT GCA GCT CTT GGT AGT AGT C (reverse)], ADAM17 [TGG TCT AGC AGA ATG TGC CC (forward), ACA CAG CCT CTT TCC AAA CC (reverse)], MMP14 [GCT ATC CTT TGC CCA CTG GT (forward), CTC CCG CTC TTC CTC AAC TC (reverse)], and GAPDH [TTC ACC ACC ATG GAG AAG GC (forward), AGG AGG CAT TGC TGA TGA TCT (reverse)].

### 2.3. Immunofluorescence Staining

The nasal polyp samples were fixed in formalin and embedded in paraffin after endoscopic sinus surgery performed under general anesthesia. The samples were obtained from 39 patients with CRSwNP (55 ± 14 years of age [mean ± SD], the ratio of male to female: 28–11). The paraffin-embedded sections were deparaffinized, rehydrated, and processed by proteinase K-induced antigen retrieval. Then, the sections were blocked and stained with anti-EpCAM (Cell signaling Technology, Danvers, MA, USA) as an epithelial cell marker, MBP (Bio-Rad, Hercules, CA, USA) as an eosinophil marker, MUC1 (Abcam) and CCL4 (Bioss, Woburn, MA, USA), followed by goat anti-rabbit CF-488 antibody or goat anti-mouse CF-647 antibody (Biotium, PA, USA). Two MUC1 antibodies were used: EP1024Y binds to the N-terminal domain and detects full-length MUC1 (MUC1-FL); EPR1023 binds to the C-terminal domain (MUC1-C) and can detect both full- and short-length MUC1 without the N-terminal domain. The intensity ratio of MUC1-FL or CCL4 to EpCAM was calculated using ImageJ (latest v. 1.54). All MUC1, CCL4, and EpCAM signals were labeled, and the intensity was evaluated using ImageJ (a single signal was reversed to 8 bits). BEAS-2B coincubated with purified eosinophils were fixed with 4% paraformaldehyde, permeabilized, and blocked. The cells were incubated with MUC1 (Abcam) and then evaluated. Control antibodies and Hoechst staining (Invitrogen, Paisley, UK) were also included in each experiment, and the slides were visualized using an FV3000 confocal microscopes (Olympus, Tokyo, Japan). The local ethics committee of Kansai Medical University approved this study (Approval number: KanIRin1313). The written informed consent was provided by all study subjects.

### 2.4. Cell Survival

To assess cell viability, the cells double stained with 7-amino-actinomycin and Annexin (BD Pharmingen, Franklin Lakes, NJ, USA) were analyzed using Aria III (BD Biosciences, Franklin Lakes, NJ, USA) and FlowJo (BD Biosciences).

### 2.5. Western Blot

The cell proteins were extracted using modified RIPA buffer (50 mM Tris HCL pH 7.4, 1.0% NP-40, 0.25% Na-deoxycholate, and 150 mM NaCl with freshly added complete protease inhibitor). The protein concentrations were defined using the BCA Protein Assay Kit (Thermo Fisher Scientific, Rockford, IL, USA) as described previously [25]. The protein extract was separated by SDS-PAGE (Bio-Rad, Hercules, CA, USA) and detected by Western blot analysis using the FUSION SOLO S imaging system (Vilber, Marne La Vallee Cedex 3, France). MUC1 was detected using rabbit monoclonal antibodies to MUC1 (EP1024Y and EPR1023, Abcam), and indicated as a ratio to β-actin.

### 2.6. Statistical Analysis

Two groups of data were compared using the paired *t*-test. Spearman’s rank method was used for calculation of correlation coefficients. ANOVA with post hoc test adjusted for multiple comparisons was used for analysis of other data, as appropriate. *p* < 0.05 was considered as statistically significant. Descriptive statistics were indicated as means ± SEM.

## 3. Results

### 3.1. MUC1-FL Expression Decreases in Epithelial Cells of Nasal Polyps from ECRS Subjects

To confirm the role of MUC1 in eosinophilic airway inflammation, the expression of MUC1 in nasal polyps obtained from patients with CRSwNP was examined. Immunofluorescence staining revealed that the expression of MUC1-FL in the epithelial cells of nasal polyps with high eosinophil count (>100/high power field [HPF]) was reduced compared with that in those with low eosinophils (<100/HPF), whereas the expression of the C-terminal domain (MUC1-C) was the same in both groups (Figure 1A). Furthermore, the expression of MUC1-FL was negatively correlated with the eosinophil count in nasal polyps (HPF) (Figure 1B), indicating that the N-terminal domain of MUC1 may be shed in eosinophilic airway inflammation.

### 3.2. Eosinophils Reduce MUC1-FL Expression in Airway Epithelial Cells

To elucidate the direct effect of eosinophils on the shedding of the MUC1 N-terminal domain, purified eosinophils were coincubated with airway epithelial cells, BEAS-2B. mRNA levels of MUC1-FL in BEAS-2B were significantly reduced by coincubation with eosinophils (Figure 2A). Further, protein levels of MUC1-FL, but not MUC1-C, in BEAS-2B were also reduced by coincubation with eosinophils (Figure 2B–D). The supernatants of eosinophilic mucin reduced the expression of MUC1-FL, but not MUC1-C, proposing that some kinds of proteins in the mucin shed the MUC1 N-terminal domain.

### 3.3. MUC1-FL Expression Negatively Correlates with CCL4 Expression in Epithelial Cells of Nasal Polyps from ECRS Subjects

CCL4 is a key cytokine released from eosinophils and epithelial cells in eosinophilic airway inflammation [26], and its level is high in mucin obtained from patients with ECRS [27]. Therefore, CCL4 in mucin might be involved in the shedding of the MUC1 N-terminal domain; accordingly, the association between CCL4 and MUC1 expression in the nasal polyps of patients with CRSwNP was examined. The epithelial cells of nasal polyps with high eosinophil count (>100/HPF) indicated a lower intensity of MUC1-FL with higher intensity of CCL4 than those with low eosinophils (<100/HPF) (Figure 3A). CCL4 expression was positively correlated with the eosinophil count in nasal polyps (HPF) (Figure 3B), which agrees with a previous report [27]. Interestingly, MUC1-FL expression was negatively correlated with CCL4 expression (Figure 3C), proposing the potential involvement of CCL4 in the N-terminal domain of MUC1 shedding in eosinophilic airway inflammation.

### 3.4. Low MUC1-FL Is Associated with Prolonged Eosinophil Survival

To examine the direct effect of CCL4 on the shedding of the N-terminal domain of MUC1, airway epithelial cells, BEAS-2B, were stimulated with CCL4. CCL4 reduced the expression of MUC1-FL (Figure 4A), concomitantly with increased expression of metalloproteases such as disintegrin and metalloprotease domain containing protein-17 (ADAM17) or matrix metalloprotease 14 (MMP14) (Figure 4B). Then, the involvement of CCL4 in the shedding of the N-terminal domain of MUC1 was confirmed using the anti-CCL4 neutralizing antibody. The anti-CCL4 neutralizing antibody restored the eosinophil-induced reduction of MUC1-FL in BEAS-2B (Figure 4B) in line with the suppression of CCL4 levels in supernatants of BEAS-2B coincubated with eosinophils (Figure 4C). Although eosinophil survival was prolonged under coincubation with BEAS-2B, the anti-CCL4 neutralizing antibody reduced the effect (Figure 4D). Notably, eosinophil survival was dose-dependently prolonged on the recombinant human MUC1-coated plate, which does not contain the N-terminal domain (Figure 4E). Altogether, eosinophil survival could be prolonged via CCL4-mediated shedding of the N-terminal domain of MUC1 in eosinophilic airway inflammation (Figure 5).

## 4. Discussion

This study showed that in eosinophilic airway inflammation, increased CCL4 is associated with the shedding of the MUC1 N-terminal domain in airway epithelial cells, leading to enhanced eosinophil survival. CCL4 is released from activated eosinophils and airway epithelial cells in type 2 inflammation [26,28]. CCL4 upregulated the expression of CD69, an eosinophil activation marker, in eosinophils and further induced CD69 upregulation even in eosinophils coincubated with BEAS-2B, proposing that CCL4 is involved in eosinophil activation. In contrast, CCL4 did not directly affect eosinophilic viability [27].

Eosinophils express several adhesion receptors such as selectins [29]. In addition, MUC1 contributes to cellular adhesive properties via selectins. However, the interaction between airway epithelial cells and eosinophils via MUC1 is unclear. Considering the findings in this study, the coexistence of eosinophils and epithelial cells induces CCL4 release from these cells, leading to the shedding of the MUC1 N-terminal domain. Moreover, eosinophils bind to the C-terminal domain including MUC1-CT, prolonging eosinophil survival. However, further studies on the mechanisms of this phenomenon by eosinophil-MUC1-CT interaction are necessary.

GRα, a key molecule for the corticosteroid response of human airway epithelial cells [30], forms a complex with MUC1-CT. This complex translocates into the nucleus and exerts an anti-inflammatory effect [22]. As we previously showed reduced GR nuclear translocation in airway epithelial cells coincubated with eosinophils [5], eosinophils could affect the GRα-MUC1-CT complex. This prolonged effect of eosinophil survival may account for the down-regulated function of GR because corticosteroids can induce eosinophil apoptosis [31]. Eosinophils in contact with epithelial cells might reduce the translocation of the GRα-MUC1-CT complex into the nucleus possibly through interaction with the MUC1 C-terminal domain after shedding the N-terminal domain, leading to reduced corticosteroid-induced apoptosis of eosinophils.

Regarding shedding, proinflammatory cytokines such as interferon-γ or tumor necrosis factor-α (TNFα) induce the dissociation of the MUC1 complex (release of MUC1 N-terminal domain from the C-terminal domain), which is catalyzed by sheddases including ADAM17 and MMPs such as MT1-MMP (MMP14) [32,33]. ADAM17 enhances leukocyte migration by activating endothelial and epithelial permeability and increases the release of inflammatory mediators from the smooth muscles and epithelial cells in the airway [34]. ADAM17 is associated with airway remodeling through enhanced extracellular matrix production and proliferation of fibroblast and epithelial cells [35,36]. MMP14, which increases in asthmatics, is involved in the aggravation of allergic airway inflammation by leukocyte recruitment and airway remodeling by upregulating the proliferation and migration of airway smooth muscle cells [37,38]. Although CCL4 enhances MMP14 expression in collaboration with TNFα in a monocytic human cell line [39], the role of CCL4 on metalloproteases expression in human airway epithelial cells is unknown. Because the stimulation of airway epithelial cell with CCL4 enhanced the expression levels of ADAM17 and MMP14, CCL4 released from eosinophils and epithelial cells might be involved in the shedding of MUC1 N-terminal domain by the increased expression of these metalloproteases. Accordingly, allergic airway inflammation might be further activated through prolonged eosinophil survival.

As a limitation of this study, we used the eosinophils from healthy individuals because it was difficult to obtain the substantial volume of patients’ blood samples required for experiments using purified eosinophils. Although the samples from healthy individuals with the lower range of eosinophils are not empirically activated, those with slightly increased eosinophils (approximately 300–500/μL) are activated during coincubation with airway epithelial cells [6]. However, further studies using the samples from patients with eosinophilic airway inflammation will be needed to replicate the disease condition and assess the credibility of this study.

## 5. Conclusions

This study presented a novel mechanism for interacting MUC1 and eosinophils in eosinophilic airway inflammation. The findings indicate a possible first step during the contact of eosinophils with airway epithelial cells. This mechanism may be valuable in developing new treatment strategies for eosinophilic airway inflammation.

## Figures and Tables

**Figure 1 cells-14-00033-f001:**
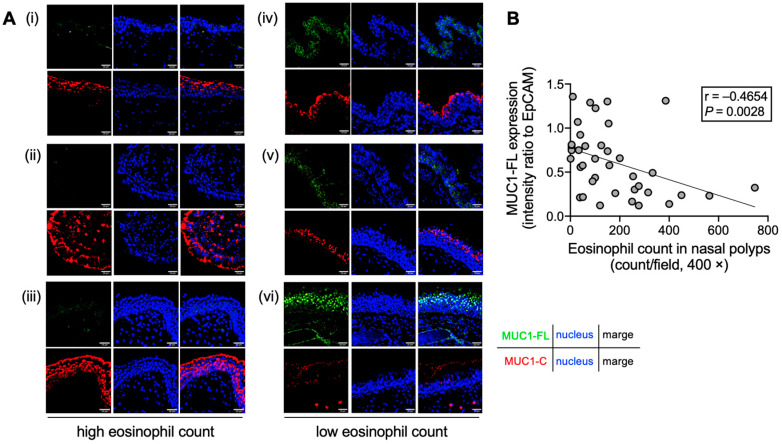
Full-length MUC1 (MUC1-FL) expression in the epithelial cells of nasal polyps. (**A**) Immunofluorescence analysis of nasal polyps obtained from patients with chronic rhinosinusitis with nasal polyps (CRSwNP) with high eosinophil count (left panels; i–iii) or those with low eosinophil count (right panels; iv–vi). MUC1-FL (green), C-terminal domain (MUC1-C, red), and the nucleus (blue) are stained. Images were captured by an FV3000 confocal microscope (400× objectives). The scale bars in the bottom-right corner indicate 10 μm. (**B**) Correlation of MUC1-FL expression with eosinophil count in nasal polyps. MUC1-FL intensity is indicated as a ratio to epithelial cell adhesion molecule (EpCAM) (n = 39).

**Figure 2 cells-14-00033-f002:**
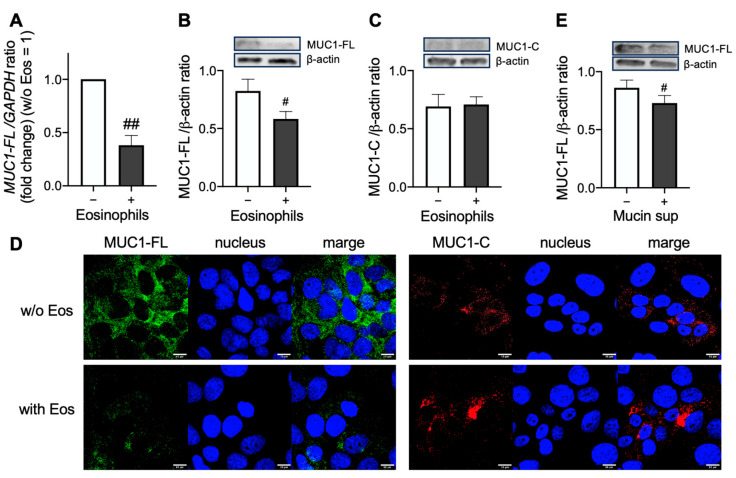
Effect of eosinophils on full-length MUC1 (MUC1-FL) expression in airway epithelial cells. (**A**–**D**) BEAS-2B cells were coincubated overnight with purified peripheral blood eosinophils. MUC1-FL mRNA levels (**A**), MUC1-FL protein levels (**B**), and MUC1 C-terminal domain (MUC1-C) protein levels (**C**) were evaluated. (**D**) Immunofluorescence analysis of MUC1-FL (green), MUC1-C (red), and the nucleus (blue) are shown in the upper (without eosinophils) and lower (with eosinophils) panels. Images were captured by an FV3000 confocal microscope (400× objectives). Scale bars in the bottom-right corner indicate 10 μm. Results were representative of at least three experiments. (**E**) MUC1-FL protein expression in BEAS-2B coincubated with the supernatants of eosinophilic mucin overnight. Patients underwent endoscopic sinus surgery under general anesthesia. Mucin samples were collected from the sinuses of refractory ECRS subjects. Values in (**A**–**C**,**E**) represent the mean ± SEM of four experiments; ^#^ *p* < 0.05, ^##^ *p* < 0.01 (vs. vehicle).

**Figure 3 cells-14-00033-f003:**
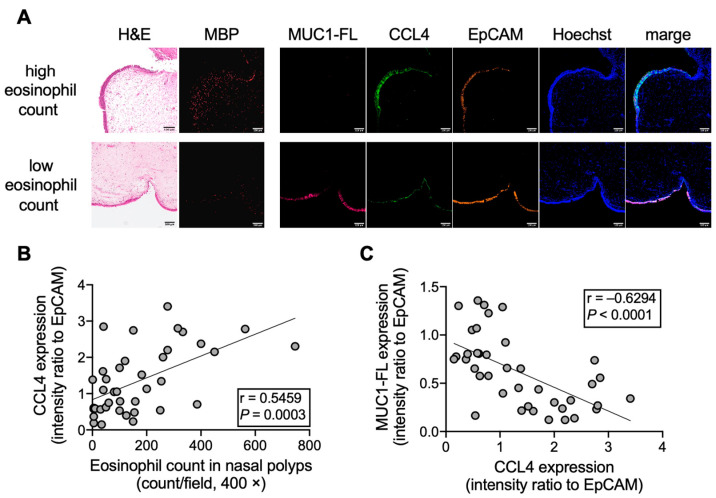
Relation between full-length MUC1 (MUC1-FL) and CCL4 expression in the epithelial cells of nasal polyps. (**A**) Immunofluorescence staining of nasal polyps obtained from patients with CRSwNP with high or low eosinophil count. MUC1-FL, CCL4, and epithelial cell adhesion molecule (EpCAM) expression levels were evaluated. MUC1-FL (pink), CCL4 (green), EpCAM (orange), MBP (red), and the nucleus (blue) are stained with hematoxylin and eosin (H&E). Images were captured by an FV3000 confocal microscope (100× objectives). The scale bars in the bottom-right corner indicate 100 μm. (**B**,**C**) Correlation of CCL4 expression with the eosinophil count in nasal polyps (**B**) and MUC1-FL expression (**C**). MUC1-FL and CCL4 intensities are indicated as a ratio to EpCAM (n = 39).

**Figure 4 cells-14-00033-f004:**
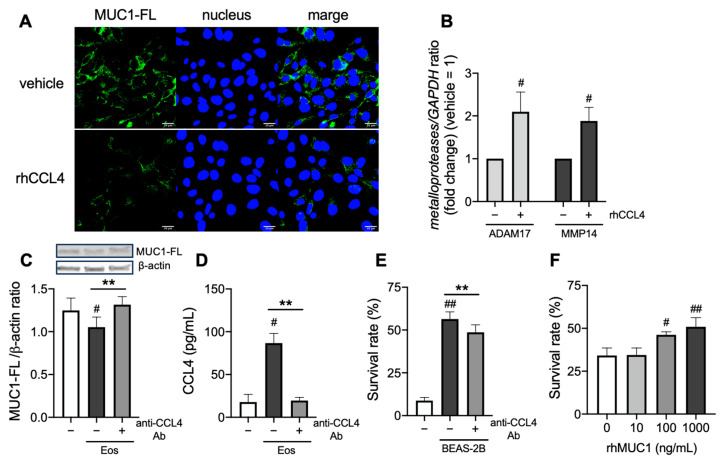
Effect of CCL4-mediated reduction of full-length MUC1 (MUC1-FL) on eosinophil survival. (**A**,**B**) BEAS-2B was stimulated overnight with recombinant human CCL4 (10 μg/mL). MUC1-FL expression (**A**) and matrix metalloproteases (ADAM17 and MMP14) mRNA levels (**B**) in BEAS-2B. (**C**–**E**) BEAS-2B and purified eosinophils were coincubated with or without anti-CCL4 neutralizing antibody (10 μg/mL). MUC1-FL protein levels in BEAS-2B (**C**), CCL4 concentration in supernatants of cell culture (**D**), and eosinophil survival (**E**) were evaluated. (**F**) Purified eosinophils were incubated overnight on a recombinant human MUC1-coated plate, followed by the evaluation of their survival. Images (MUC1-FL, green; nucleus, blue) in A were captured by an FV3000 confocal microscope (400× objectives) with scale bars (20 μm) in the bottom-right corner, which were representative of at least three experiments. The values in (**B**–**F**) represent the mean ± SEM of four experiments. ^#^ *p* < 0.05, ^##^ *p* < 0.01 (vs. without rhCCL4 in (**B**), without eosinophils in (**C**,**D**), without BEAS-2B in (**E**), and without rhMUC1 in (**F**)). ** *p* < 0.01 (between the two groups).

**Figure 5 cells-14-00033-f005:**
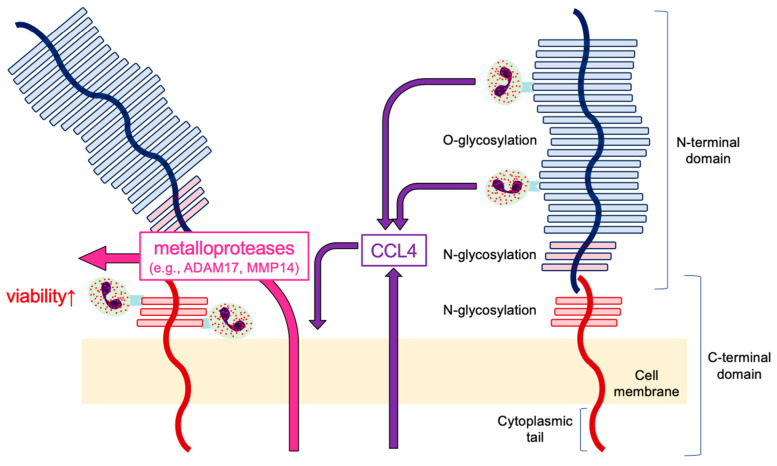
Mechanism of prolonged eosinophil survival associated with MUC1 in airway epithelial cells. Eosinophils–airway epithelial cells contact activates both cells and induces CCL4 release from them. CCL4 might be involved in the shedding of the MUC1 N-terminal domain by increased expression of these metalloproteases (e.g., ADAM17 and MMP14). The viability of eosinophils bound to the MUC1 C-terminal domain could be upregulated.

## Data Availability

The original contributions presented in this study are included in the article. Further inquiries can be directed to the corresponding author.

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
