# Peer review of "CCL4 Affects Eosinophil Survival via the Shedding of the MUC1 N-Terminal Domain in Airway Inflammation"

_cells, 2024, doi:10.3390/cells14010033_

Round 1

Reviewer 1 Report

Comments and Suggestions for Authors

The work is focused on a particular and relatively underestimated segment of eosinophilic inflammation, namely the extracellular barriers. Recently the search for new therapeutic approaches enlightens this segment. The manuscript presents a very precise and well-designed study with promising results. As a minor remark, I would say that in the clinical setting, eosinophilia is defined as an absolute number (above 500 /mm3), not as % of WBC. (line 76). In my opinion, the manuscript has very important input in diagnosing and possibly treating eosinophilic airway inflammation.

Author Response

Reviewer #1:

We would like to thank for Reviewer’s favorable comments. We tried to answer the questions.

Comments:

  1. As a minor remark, I would say that in the clinical setting, eosinophilia is defined as an absolute number (above 500 /mm3), not as % of WBC. (line 76). In my opinion, the manuscript has very important input in diagnosing and possibly treating eosinophilic airway inflammation.

(Response)

According to reviewer’s suggestions, we changed it to “slightly increased eosinophils (approximately 300–500 /mL)”.

Reviewer 2 Report

Comments and Suggestions for Authors

By using nasal polyp samples from patients suffering from ECRS and eosinophils and BEAS-2B The authors demonstrated that CCL4 affects eosinophil survival via the shedding of MUC1 N-2 terminal domain in airway inflammation.

Some suggestions:

1.       The study involved patient samples, but the materials and methods state no corresponding ethical approval. The number of patients involved in the study and other details (age, sex, etc.) should also be stated in the materials and method section.

2.       The antibodies used and protocols in the Western blot should be stated in the method section.

3.       The co-culture protocols of eosinophils and BEAS-2B were also missing.

4.       The authors use the eosinophils from healthy individuals but it is difficult to interpolate to the disease condition in ECRS, the authors should justify the rationale and limitation in the discussion.

5.       Is there any appropriate animal models to further validate the current findings?

Author Response

Reviewer #2:

We would like to thank for Reviewer’s favorable comments. We tried to answer the questions.

Comments:

  1.        The study involved patient samples, but the materials and methods state no corresponding ethical approval. The number of patients involved in the study and other details (age, sex, etc.) should also be stated in the materials and method section.

(Response)

We mentioned about ethical approval in 2.3 (replaced from 2.1) as follows;

“This study was approved by the local ethics committee of Kansai Medical University (Approval number: KanIRin1313). All study participants provided written informed consent.”

Further, we added some information about the patients in 2.3 as follow;

“The samples were obtained from 39 patients with CRSwNP (55 ± 14 years of age [mean ± SD], the ratio of male to female: 28-11).”

  1. The antibodies used and protocols in the Western blot should be stated in the method section.

(Response)

We already mentioned about them in 2.5. Please check again.

  1. The co-culture protocols of eosinophils and BEAS-2B were also missing.

(Response)

We added the brief protocol in 2.1 as follows;

“Briefly, BEAS-2B cells seeded in a cell culture plate one day before were stimulated with recombinant human CCL4 overnight (for 20–24 h) or were coincubated with purified eosinophils in the presence or absence of anti-CCL4 antibody overnight.”

  1. The authors use the eosinophils from healthy individuals but it is difficult to interpolate to the disease condition in ECRS, the authors should justify the rationale and limitation in the discussion.

(Response)

As reviewer pointed out, we discussed about this issue as a limitation in the discussion as follows;

As a limitation of this study, we used the eosinophils from healthy individuals because it was difficult to obtain substantial patients’ blood samples required for experiments using purified eosinophils. Although the samples from healthy individuals with the lower range of eosinophils are not empirically activated, those with slightly increased eosinophils (approximately 300–500 /mL) are activated during coincubation with airway epithelial cells [1]. However, further study using the samples from patients with eosinophilic airway inflammation will be needed to replicate the desease condition and assess the credibility of this study.

  1. Is there any appropriate animal models to further validate the current findings?

(Response)

There are some animal models with MUC1 deficiency in which nasal epithelial barrier impaires under allergic condtion [2, 3]. However, in these model total MUC1, not only the N-terminal domain but also the C-terminal domain is deficient. Therefore, there are no appropriate animal models for further validation of our findings so far.

  1. Kobayashi, Y., H. Yasuba, M. Asako, T. Yamamoto, H. Takano, K. Tomoda, A. Kanda, and H. Iwai. "Hfa-Bdp Metered-Dose Inhaler Exhaled through the Nose Improves Eosinophilic Chronic Rhinosinusitis with Bronchial Asthma: A Blinded, Placebo-Controlled Study." Front Immunol 9 (2018): 2192.
  2. Zhang, C., Y. Wang, W. Liao, T. Liang, W. Liu, J. Xie, X. Wang, P. Yang, W. Lu, and X. Zhang. "Muc1 Deficiency Induces the Nasal Epithelial Barrier Dysfunction Via Rbfox3 Shortage Augment Ubiquitin-Proteasomal Degradation in Allergic Rhinitis Pathogenesis." Allergy 77, no. 5 (2022): 1596-99.
  3. Zhou, L. B., Y. M. Zheng, W. J. Liao, L. J. Song, X. Meng, X. Gong, G. Chen, W. X. Liu, Y. Q. Wang, D. M. Han, N. S. Zhong, W. J. Lu, P. C. Yang, and X. W. Zhang. "Muc1 Deficiency Promotes Nasal Epithelial Barrier Dysfunction in Subjects with Allergic Rhinitis." J Allergy Clin Immunol 144, no. 6 (2019): 1716-19.e5.

Reviewer 3 Report

Comments and Suggestions for Authors

This study showed that in eosinophilic airway inflammation, increased CCL4 is associated with the shedding of MUC1 N-terminal domain in airway epithelial cells, leading to enhanced eosinophil survival. CCL4 is released from not only activated eosinophils but also airway epithelial cells in type 2 inflammation.

This is an interesting study with a clear methodology and experimental approach in order to reach the objectives of the study. Nevertheless, the introduction about the Mucin glycoproteins is a little confuse my recommendation is to put a table or a figure in order to clarify the mucins interactions and sites of actions for settle the research question

Author Response

Reviewer #3:

We would like to thank for Reviewer’s favorable comments. We tried to answer the questions.

Comments:

  1. Nevertheless, the introduction about the Mucin glycoproteins is a little confuse my recommendation is to put a table or a figure in order to clarify the mucins interactions and sites of actions for settle the research question

(Response)

According to reviewer’s suggestions, we added some information as follows;

“The N-terminal extracellular domain on airway epithelial cells acts as a contact point to pathogens or adhesion molecules in other cells [1, 2].”

“We hypothesized that eosinophils first contact the N-terminal extracellular domain of MUC1 on airway epithelial cells and then exert some influence on airway inflammation (see the right side of Figure 5).”

  1. McAuley, J. L., L. Corcilius, H. X. Tan, R. J. Payne, M. A. McGuckin, and L. E. Brown. "The Cell Surface Mucin Muc1 Limits the Severity of Influenza a Virus Infection." Mucosal Immunol 10, no. 6 (2017): 1581-93.
  2. Rahn, J. J., Q. Shen, B. K. Mah, and J. C. Hugh. "Muc1 Initiates a Calcium Signal after Ligation by Intercellular Adhesion Molecule-1." J Biol Chem 279, no. 28 (2004): 29386-90.